# The Impact of Sintering Technology and Milling Technology on Fitting Titanium Crowns to Abutment Teeth—In Vitro Studies

**DOI:** 10.3390/ma15175835

**Published:** 2022-08-24

**Authors:** Wojciech Ryniewicz, Łukasz Bojko, Anna M. Ryniewicz

**Affiliations:** 1Department of Dental Prosthodontics and Orthodontics, Dental Institute, Faculty of Medicine, Jagiellonian University Medical College, 4 Montelupich Street, 31-155 Krakow, Poland; 2Faculty of Mechanical Engineering and Robotics, AGH University of Science and Technology, 30 Mickiewicza Ave., 30-059 Krakow, Poland

**Keywords:** prosthetics, SLM, milling, marginal fit, internal fit

## Abstract

Introduction: The aim of the study is to evaluate the marginal and internal fit of titanium alloy (Ti6Al4V) crowns using the Selective Laser Melting (SLM) method and CAD/CAM milling. Materials and Methods: The research materials are abutment teeth and prosthetic crowns. The method is based on scanning the abutments and the interior of the substructures, creating their 3D models, using the program for comparison, and determining error maps of fitting crowns to the reference models, in the form of positive and negative deviations. Adding the deviations gives information about the tightness of the crowns. The Shapiro–Wilk test and the one-way ANOVA analysis were performed. The level of significance was *p* = 0.05. Results: The crowns made in SLM, a slightly better internal fit was found than for milled crowns, as well as a comparable marginal fit. The mean deviations for the sintering were the values [mm]: −0.039 and +0.107 for tooth 15 and −0.033 and +0.091 for tooth 36, and for the milling –0.048 and +0.110 for tooth 15 and –0.038 and +0.096 and for tooth 36. Conclusion: Based on the research conducted and the experience in therapeutic procedures, it can be indicated that the fitting of titanium alloy crowns in SLM and milling meets the clinical requirements. To evaluate the technology, a method was developed that determines the accuracy of mapping the shape of the tooth abutments in the crown substructures for the individual conditions of the patient.

## 1. Introduction

Fixed prosthetic restoration should meet the following: health-related requirements in terms of biocompatible protection of the tissues of the oral cavity and proper functioning of the patient’s organs; mechanical requirements for strength (protection of tissue strain, temporary and fatigue parameters of the structure) and aesthetic requirements related to the location of the gingival margin, as well as shape, thickness and color of the veneer [1,2].

The connection between a cemented crown and a tooth is always a potential leakage area. The more precisely the restoration is fitted to the abutment tooth, the lower the probability of secondary caries and periodontal disease [3,4,5]. Proper shaping of the chamfer on the abutment tooth, precise mapping of the shape in the supporting substructure, and ensuring crown retention in clinical procedures determine the long-term prognosis of treatment [6,7,8]. The currently implemented prosthetic treatment process involves many changes related to the use of digital methods. They are used in imaging and diagnostics. In digital procedures, the design and modeling of the restoration structure, as well as its programmed production by milling or printing, are performed. The method proposed in the study ensures a precise evaluation of the accuracy of the manufactured structure. The evaluation is based on digital control of the fit of the interior of the substructure to the abutments of the teeth. The compatibility of the developed digital objects enables a much better supply to the prosthetic patient. The doctor has the possibility of biomechanical virtual stimulation of the masticatory organ and determining the spatial distribution of stresses and displacements in the tissues of the stomatognathic system (SS) and in the construction used, indicating the zones of physiological tissue stress and the evaluation of the strength of the restoration.

Due to the material characteristics of titanium and its alloys (biocompatibility, resistance to corrosion and electrochemical degradation, lightness, low thermal conductivity, low allergic impact, low Young’s modulus, and high mechanical strength), it is currently the most preferred metal for prosthetic and implant–prosthetic structures [9,10,11].

The crowns in question are layered prostheses. The outer veneering layer is made of a dedicated aesthetic glass ceramic or hybrid ceramic, and the substructure is made of a titanium alloy with aluminum and vanadium. An alternative to using crowns on a titanium substructure are all-ceramic crowns or crowns on a cobalt–chrome substructure. The indications for the use of crowns on a titanium substructure is, in strictly defined cases, strength considerations, allergies to other metals used on the substructures (nickel, palladium, chromium, cobalt) and the possibility of galvanic cells. Low oral pH can lead to chemical corrosion. In long-term observations, reports in the literature confirm the low failure rate associated with the use of crowns and bridges on a titanium base [2,12,13,14,15,16,17,18,19]. Initially, titanium was used for the substructures in a technically clean condition (CAD/CAM Kavo Everest procedure). The milling process was difficult due to the high resistance to machining [20]. The analyzed titanium alloy Ti6Al4V has a dental approval certificate. In the process of producing blocks and in the milling process, it gives a satisfactory phase and structural homogeneity, stability, and corrosion resistance.

The limitation of the use of titanium biomaterials is caused by the difficulty of processing titanium under laboratory conditions. High affinity to oxygen, nitrogen, carbon, hydrogen, silicon, low specific gravity, and high melting point make the process of casting this metal difficult and problematic. Due to the many difficulties of casting titanium, it is necessary to carry out this process in a noble gas blanket and use a large centrifugal force. This technology is being replaced by digital CAD/CAM methods: Selective Laser Melting (SLM) and milling from factory blocks. Thanks to the use of digital methods, an alternative to traditional methods has been created [21,22].

Many researchers deal with the problem of the accuracy of mapping the shape of prosthetic abutments in the substructure, which is the basis for the final optimal crown fabrication, primarily in CAD/CAM systems [23,24,25,26]. This problem concerns the fabrication of various construction solutions for prosthetic crowns. It has been analyzed in ceramic crowns [3], metal crowns veneered with ceramics [4,22,27,28] or ceramic crowns made of zirconium oxide stabilized with yttrium [29,30,31,32]. Methods to evaluate the accuracy of abutments and crowns mapping rely on direct measurements of the interior of the substructure with the use of a wax impression and measurements of the prosthetic abutment [23,33], use the technique of scanning replicas and reference models [23,34,35], shape analysis with the use of structured light [33,36], and crown fit analysis using CT or micro-CT [34,37,38].

The verification of the fit of the crown to the abutment tooth and the accuracy of the shape dimensional depending on the technology of its production is a difficult measurement problem [5,22,27]. Therefore, for titanium alloy crowns, an in vitro experiment was planned with the observance of the obligatory clinical rules and the experience resulting from prosthetic practice [20,39,40,41].

The aim of the study is to evaluate the marginal and internal fit of single crown fixed dentures made of titanium–aluminum–vanadium alloy (Ti6Al4V) made in two CAD/CAM technologies: SLM and milling. This is done with a newly developed method based on digital techniques. Using this method, the accuracy of mapping the shape of crowns in relation to clinical prosthetic abutments can be evaluated and compared.

## 2. Materials and Methods

The research material consists of prosthetic crowns made of Ti6Al4V alloy for the second premolar tooth on the right side (tooth 15) and the first molar tooth on the left side (tooth 36), made of 5 for the premolar and 5 for the molar (Figure 1). On each ground abutment tooth, crowns were made in both technologies: SLM and milling. The research program was approved by the Jagiellonian University Bioethics Committee (Approval No. KBET 122.6120.18.2016). Informed consent was obtained from all subjects involved in the study.

The premolar was ground on 5 toothed phantoms of the maxilla, and the molar was ground on 5 toothed phantoms of the mandible. KaVo phantoms teeth were used. The crowns of the molars and premolars were ground with a full obtuse degree of 140° ± 4° according to current clinical criteria. Folding master casts of the full dental arch were made with the use of superhard type IV plaster—Fujirock EP Pastel Yellow, by Fuji, on the basis of one-time, two-layer impressions. The impressions were made with a silicone addition mass with a soft putty consistency of Bisico S1 soft by Bisico, as well as a complementary impression made with a super hydrophilic, light-bodied mass Bisico S4 Suhy from Bisico [33,39,40]. Reference models had abutment teeth covered with a powder for precise scanning, which were separated from the master cast. The research method is presented in the diagram (Figure 2). 

The scanning was carried out using a D750 scanner equipped with a blue laser and two cameras with a resolution of 1.3 Mpix—the 3Shape Dental System 2016 by 3Shape (Figure 3). The accuracy of the measurements was 10 µm, and the spatial resolution was 0.01 mm × 0.01 mm × 0.01 mm. In the CAD design procedure of the reference models, a place was secured for the material that binds the prosthetic crown to the abutment (Figure 4).

Laboratory procedures for the fabrication of the substructure began with scanning of the master cast with marking of the grinding margin (margin line) on the virtual models. Then, the path of introducing the base course was determined, eliminating the undercuts. In the next stage, the substructures were designed, securing a cement place (cement gap) in the chamfer zone—0.045 mm and in the internal zone—0.085 mm (Figure 5).

Based on the STL files obtained, 5 SLM crowns (Ti6Al4V ELI-0406 powder, Renishaw, Warsaw, Poland) and 5 milled crowns (Ti6Al4V-Starbond Ti5 Disc alloy, Scheftner, Mainz, Germany) were created for teeth 15 and 36, respectively (Figure 1). The SLM uses a titanium alloy powder for medical and dental applications in accordance with document H-5983-9026. The milling uses disks used for the production of dental prosthetic elements, made of Ti6Al4V grade 5 ELI titanium alloy according to ISO 5832-2 and ASTM F136. The inner shape of the crowns was scanned using the same D750 scanner (3Shape, Copenhagen, Denmark). In this way, 20 spatial test models were created. The spatial resolution of the model grid was: 0.1 mm × 0.1 mm × 0.1 mm. For premolars 15, point clouds were obtained in the range of 7089–7139 points 3D, and for molars 36 point clouds in the range of 10,937–11,304 points 3D. In the research, the fit of Ti6Al4V crowns made in the technologies tested was compared to the reference models. The best-fit method was used for the fit analysis using the Geomagic Qualify 12 program. In this program, in the best-fitting procedure, using the least-squares method, the surfaces to be compared are automatically positioned, and the positive and negative deviations resulting from the accuracy of mapping the shape of the test crown model relative to the reference model are identified. The shape errors of the chamfer zone determining the marginal fit and errors in fitting the inner surfaces of the crowns to the abutments were identified in the form of global fit maps. The distributions of the alignment deviations in the vertical sections were also determined with the buccal-palatal planes γ1, γ2 and γ3 perpendicular to the maxillary dental arch and the buccal-lingual planes ε1, ε2 and ε3 perpendicular to the dental arch of the mandible (Figure 6). Sections were made in the mesial–distal direction. The errors in fitting the substructure to the abutment were determined by the deviation values in the areas analyzed. Information on the tightness of the substructure in relation to the abutment can be obtained by summing the absolute values of the deviations at the nodal points. The greater the deviations, the less precisely the crowns have been made, which during use, especially in chewing conditions, can cause cement cracking and deterioration of tightness.

## 3. Results

In the research procedures, an analysis of the marginal and internal fit of the crowns made of Ti6Al4V alloy with SLM and the milling to the appropriate models of the abutment teeth, constituting reference models, was performed. Selected error maps of the crown shape mapping in relation to the abutments for premolars and molars are presented (Figure 7, Figure 8, Figure 9 and Figure 10). The program enables the presentation of these maps in any isometric location. The errors for all crowns tested for both technologies are in the range: −0.178 mm ÷ 0.247 mm.

The errors in mapping the shape of crowns made of Ti6Al4V alloy with SLM are in the range: −0.178 mm ÷ 0.200 mm for the premolar and −0.138 mm ÷ 0.246 mm for the molar, and from the milling in the range: −0.143 mm ÷ 0.200 mm for the premolar and from −0.168 mm to 0.247 mm for a molar. The analyses of the reports, in the form of summary lists and maps, show that all maps and histograms are characterized by small mean values of negative and positive shape deviations (Table 1).

Asymmetric distributions were observed for both SLM and milling (Figure 11). For a comparative analysis of the accuracy of crown shape mapping in relation to abutments, depending on the manufacturing technology, zero deviations in the range of ±0.02 mm were adopted. In SLM, the zero deviation amounts to 22.38% for the premolar and 26.13% for the molar. In the case of milling, zero deviations constitute 16.98% for the premolar and 25.24% for the molar.

In crowns made of Ti6Al4V alloy, slightly better shape mapping was observed in SLM than in milling, as well as lower values of the average positive and negative deviations, and a higher percentage of zero deviations in the mapping of the shape of the crowns to the molar tooth than to the premolar tooth.

The analysis of a wider range of deviations in shape mapping, from −0.043 mm to 0.043 mm in crowns made of Ti6Al4V alloy, from SLM shows that 42.06% of the deviations for the premolar tooth and 46.51% for the molar tooth were in the mentioned range. In milling, a lower percentage of deviations in this range was observed for both the premolar 34.18% and the molar 43.64%. In the case of a molar crown, a more precise mapping of the shape of the prosthetic abutment was found.

Based on histograms of the distribution of the accuracy deviations of the mapping in crowns for premolars and molars, produced in both technologies, it can be concluded that the largest percentage is represented by zero deviations.

The analysis of the accuracy of mapping the shape of crowns made of Ti6Al4V alloy, in relation to the abutments for the tested technologies, was extended by the procedure of making cross sections in vertical planes perpendicular to the dental arches. For premolars, cross sections were made with buccal–palatal planes perpendicular to the maxillary dental arch γ1, γ2 and γ3 (Figure 12 and Figure 13). For molars, cross-sections were made with the buccal–lingual planes perpendicular to the dental arch of the mandible ε1, ε2 and ε3 (Figure 14 and Figure 15). The distributions of the deviations in the accuracy of mapping the shape of the crowns in the premolars and molars in all three sections confirm that the tightness in both technologies was at a satisfactory level, from −0.13 mm to 0.16 mm. The manufacture of Ti6Al4V crowns guaranteed a marginal fit in the chamfer zone and an internal fit in the area of the occlusal and lateral surfaces. The fit of the contact occlusal surfaces of the crowns on the premolar was comparable: in SLM from −0.08 mm to 0.01 mm, and in the case of milling: from −0.08 mm to −0.02 mm. In the case of a molar, a slightly better fit in the area of the occlusal surfaces occurred in milling: from −0.04 mm to 0.02 mm, and in SLM, it ranged from −0.05 mm to 0.06 mm. The fitting of the side walls of the crowns on the premolar was comparable for both technologies, from 0.02 mm to 0.07 mm, and in the case of the molar, slightly better mapping was observed for milling: from 0.03 mm to 0.15 mm than for SLM: from 0.02 mm to 0.16 mm. There were negative deviations in the marginal fit of the SLM and the milling crowns. Shape mapping errors in the crown chamfer area on the premolar tooth in milling ranged from −0.07 mm to −0.04 mm and were slightly smaller than for SLM, where the deviations of shape accuracy were in the range: from 0.08 mm to –0.01 mm. A similar situation occurred with the molar. A better marginal fit can be found for milling in the range: from −0.09 mm to −0.01 mm than for SLM: from −0.13 mm to −0.04 mm.

The results obtained were analyzed using the Statistica program. In the first stage, the results were analyzed using descriptive statistics. The distributions of the individual variables were also examined using the Shapiro–Wilk test, presenting them in graphical and numerical form. The normality of variable distributions for premolars and molars was investigated in each group after adopting the null hypothesis of H_0_. This hypothesis assumes that the deviation distribution is a normal distribution. It was assumed that the value of the significance level was assumed to be α = 0.05. The analysis result, based on the Shapiro–Wilk test, was compared with the adopted significance level α. In the case when *p* < α we reject H_0_ assuming H_1_, and when *p* > α—there are no grounds to reject the H_0_ hypothesis. The results of the Shapiro–Wilk tests were always greater than 0.05. Therefore, there are no grounds to reject the H_0_ hypothesis, i.e., the evaluated distributions are normal distributions. For this comparison, the significance level was α = 0.05. H_0_: There are no statistically significant differences between the group distribution of premolars and molars. H_1_: there is a statistically significant difference between the group distribution of premolars and molars. In the analyzed procedure, *p* > α, i.e., in the group of premolars and molars there are no statistically significant differences. Then, tests were carried out using one-way ANOVA. To perform the analysis, the following criteria had to be met: normality of the distribution, homogeneity of variance, and equality of the studied groups. The H_0_ hypothesis assumes that all groups are equal and that there are no differences between the SLM and the milling groups, and the H_1_ hypothesis: there is a difference between the SLM and the milling groups. The result of this test was *p* > 0.05. The H_0_ hypothesis was maintained. If there are no differences, then there is no need to perform post hoc tests.

## 4. Discussion

The accuracy of mapping the shape of crowns is a basic criterion to ensure optimal tightness and then adhesion between the abutment and the supporting structure. In the biomechanical aspect, in chewing conditions, it is particularly important in areas where there are no natural side support zones and where the highest stress accumulation occurs [41].

The team also conducted studies on the accuracy of mapping the shape of CoCrMo crowns in relation to prosthetic abutments [27]. Research covered the technologies: SLM and milling and concerned teeth 15 and 36. The results of experiments on cobalt alloys and titanium alloys were compared. In crowns made of Ti6Al4V alloy, a lower percentage of deviations can be observed, confirming the very good mapping of the reference model shape in relation to the crowns in both teeth. The worse mapping of the crown shape in relation to the abutment made of Ti6Al4V alloys, compared to the mapping made of CoCrMo alloys, may be caused by a more difficult procedure in the machining of titanium alloys.

The applied Geomagic Qualify 12 program is a reliable tool for the geometric analysis and evaluation of clinically developed tooth prosthetic abutments and crowns made on these teeth. The development of the abutment involves creating a chamfer in the periphery of the tooth with a constant opening angle in a tissue-sparing procedure and in the proper preparation of the gingival zone. This preparation aims to create conditions for embedding the crown under development, while maintaining appropriate tightness and retention, and protecting the occurrence of stress accumulation zones in chewing and occlusion.

The evaluated method is based on:A contactless measurement procedure with a high-resolution scanner;Modeling the abutment with particular emphasis on the preparation zone;Modeling the crown in the internal and marginal area with particular emphasis on the chamfer area;Using a software tool to determine, qualitatively and quantitatively, errors in fitting the interior of the crown to the reference abutment model.

Dental abutments can be scanned directly in the mouth or using impression material mapping [42]. The use of impression materials is a certain limitation of the study in clinical procedures. As part of the work, our team evaluated the accuracy of mapping the prosthetic substructure with the use of digital impressions and mapping using analog methods. There were no statistically significant differences in the mapping with digital and analog methods. Due to much more work performed with analog methods and many years of distant observations, a decision was made to use impressions with prosthetic masses and scan models. Research was carried out in vitro, so mapping of the prosthetic substructure with the use of masses was not associated with any inconvenience to the patient.

The accuracy of mapping the shape of crowns can be evaluated by the tightness or fitting of the structure. For effective long-term therapy with the use of fixed restorations, it is necessary to achieve structure tightness [28,29,33,43]. The lack of clinically appropriate marginal tightness may result in cement washing, which is associated with biological complications, such as pulpitis, periodontal problems, and secondary caries [36,37]. Uneven internal gaps may cause cement chipping due to increased contact stresses under chewing conditions. This situation can lead to a leaky crown and its loosening. Undoubtedly, the most important area is the marginal fit. This is indicated by FEM simulation studies, in which, under chewing conditions, stress distribution maps were determined in the model, including the mandible, abutment teeth, and the construction of the double-banked bridge [41]. There is a significant variation in the distribution and stress values. The maximum stress values are located in the marginal areas of the crowns and chamfers of the abutments, making them more prone to loss of tightness due to errors in fitting. At the same time, the marginal areas are in contact with saliva and are most vulnerable to bacterial penetration and plaque build-up.

The tightness in different tests is defined differently. Furthermore, various techniques are used to measure internal and marginal gaps [23,37,44,45]. Along with the increasing number of design methods in the CAD/CAM system, numerical methods of evaluating the accuracy of mapping the internal shape of the crown in relation to the tooth abutment have become of particular importance in reconstructive dentistry.

The literature evaluates fit or tightness in different technologies. Marginal tightness in the chamfer zone in SLM is reported in the range of 43–108 µm, and the internal tightness in the side wall zone is indicated in the range of 82–415 µm [46,47]. According to the literature data, the marginal tightness in the chamfer zone in milling is in the range of 49–129 µm, and the internal tightness is in the range of 95–377 µm [47]. In casting, the marginal tightness in the chamfer zone is given in the range of 55–127 µm, and the internal tightness is given in the range of 79–360 µm [46]. The research results presented in the study were confirmed in studies conducted at Shanghai Jiao Tong University and the University of Alabama in Birmingham and fell within the ranges reported in the literature.

The methods discussed in the Introduction do not use an impartial tool–shape-fit evaluation in an automatic numerical procedure free of errors related to the person carrying out the measurement. The essence of the experiment is the spatial analysis of fit and tightness, because it is accurate and reliable for the long-term clinical prognosis.

The application of digital methods for treatment and prosthetic supplies enables control, comparison, and evaluation at subsequent stages of diagnostics, design, and production, which increase the accuracy of the solutions performed. It allows the use of new biomaterials and new technologies. CAD/CAM systems have been present for over 20 years. This technology continues to develop and enables the performance of new types of prosthetic work.

Thanks to the method used, the accuracy of mapping the shape of prosthetic abutments in the substructures on the premolar and molar crowns, dedicated to individual conditions, and produced in the following technologies: SLM and milling were evaluated.

A better mapping of the shape of the crown in relation to the abutment was observed in the case of a molar than for a premolar.

For crowns made of Ti6Al4V alloy in SLM, a slightly better internal fit was observed than for CAD/CAM milled crowns, was observed, as well as a comparable marginal fit.

Based on the research conducted, it can be concluded that the tightness of the prosthetic crowns made of Ti6Al4V alloy in SLM and milling meets the clinical requirements.

The results of this work may be useful in a clinical setting. They allow researchers to evaluate how the technology of producing a substructure made of the same biomaterial affects the accuracy of fitting crowns to the tooth abutments. It can also be noticed that titanium milling has a less favorable effect on marginal fit, which may be the result of the impact of the strength parameters on accuracy in machining.

Taking into account the destruction of tools and very large material losses in milling, aside from the higher costs of the device for making the structure in SLM, laser melting technology from selective metal powders can be considered prospective.

## 5. Conclusions

The developed method of the accuracy of mapping the shape of crowns in relation to clinical prosthetic abutments allowed the evaluation and comparison of the constructions made in digital technologies.For Ti6Al4V crowns made with SLM, a slightly better fit was observed in the internal and marginal areas than in the case of milling.Regardless of the material and manufacturing technology, a better mapping of the crown shape in relation to the abutment was observed in the case of a molar than for a premolar.Based on the conducted tests, it can be concluded that the tightness of the prosthetic crowns made in the CAD/CAM system from Ti6Al4V alloy in SLM and the milling meets the clinical requirements.The implemented methodology for the evaluation of prosthetic crowns allows the use of test results in the supply procedures of individual patients in terms of metrological parameters and biomechanical conditions.

## Figures and Tables

**Figure 1 materials-15-05835-f001:**
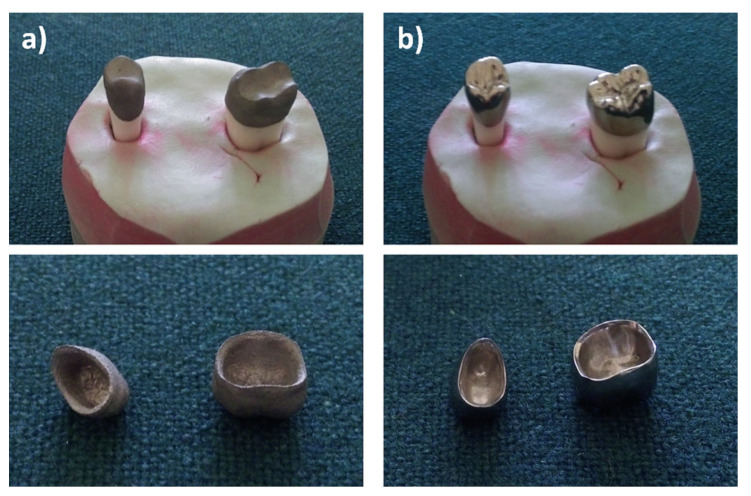
Reference models and crowns made of Ti6Al4V alloy for premolars and molars produced in the following technologies: (**a**) SLM, and (**b**) milling.

**Figure 2 materials-15-05835-f002:**
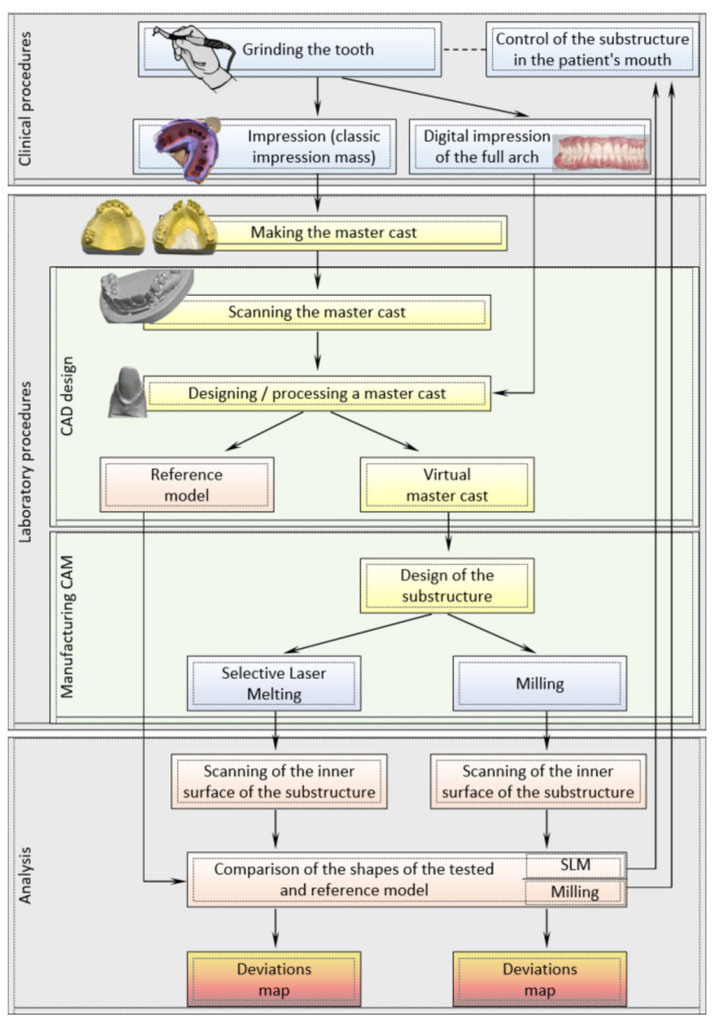
The scheme of the research method.

**Figure 3 materials-15-05835-f003:**
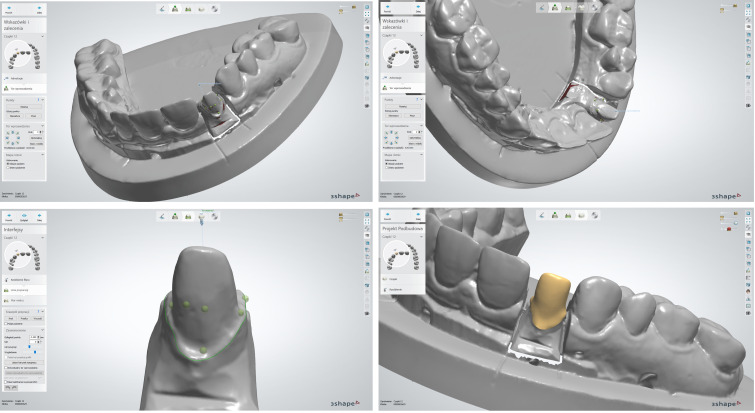
Abutment scanning and substructure design in the 3Shape Dental System clinical system.

**Figure 4 materials-15-05835-f004:**
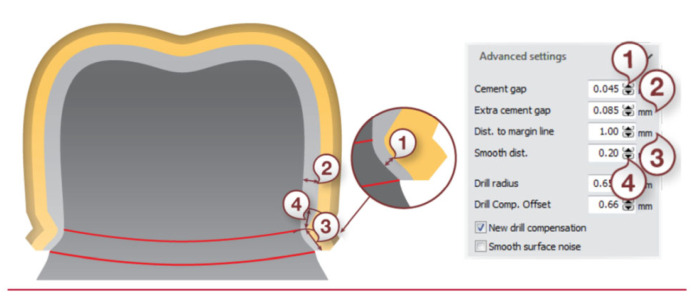
Design parameters: (1) cement gap, (2) extra cement gap, (3) distance to margin line, and (4) smooth distance (3Shape Dental System 2016 User Manual).

**Figure 5 materials-15-05835-f005:**
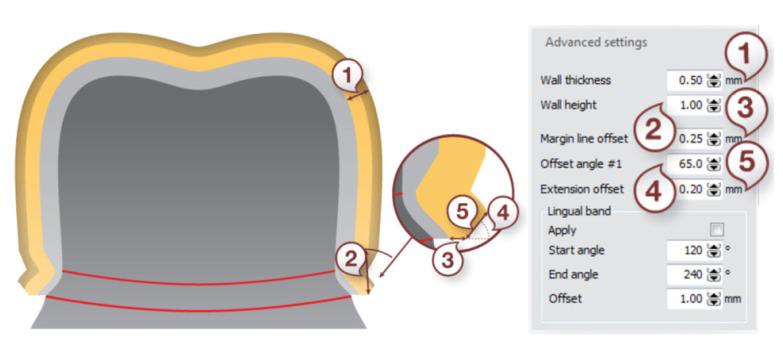
Design parameters: (1) wall thickness, (2) wall height, (3) margin line offset, (4) offset angle #1, and (5) extension offset (3Shape Dental System 2016 User Manual).

**Figure 6 materials-15-05835-f006:**
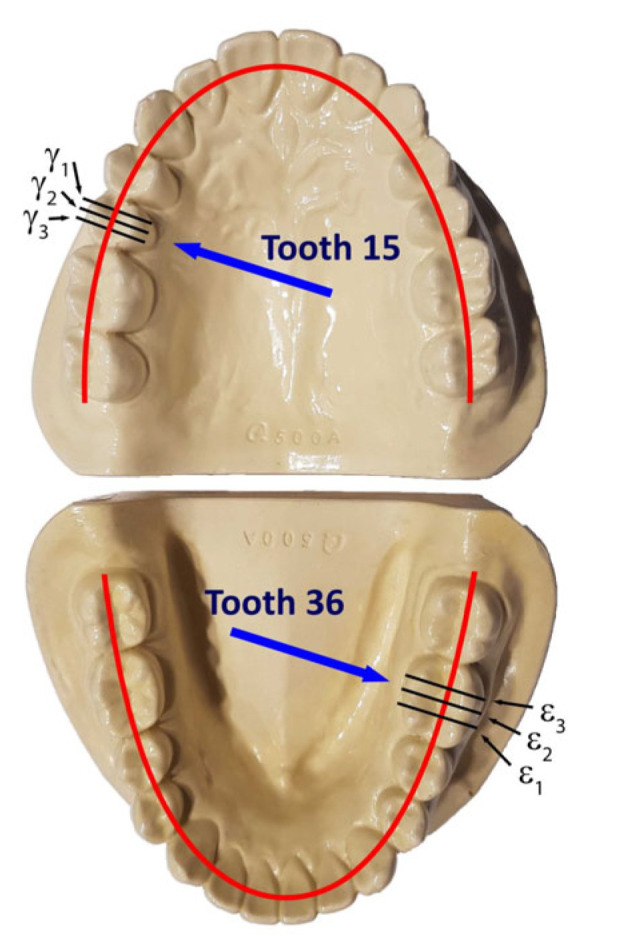
The location of the teeth in the view of the maxillary and mandibular arches with the indication of the cross-section planes for the analysis of shape deviations.

**Figure 7 materials-15-05835-f007:**
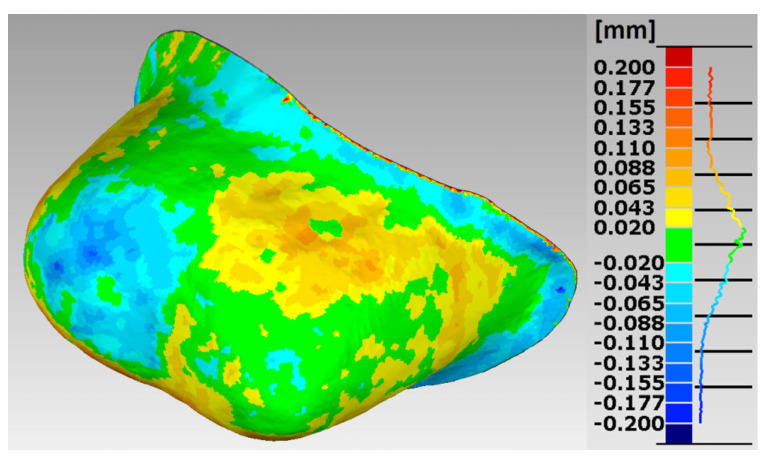
Map of errors in mapping the shape of a crown made of Ti6Al4V alloy in SLM for premolar 15.

**Figure 8 materials-15-05835-f008:**
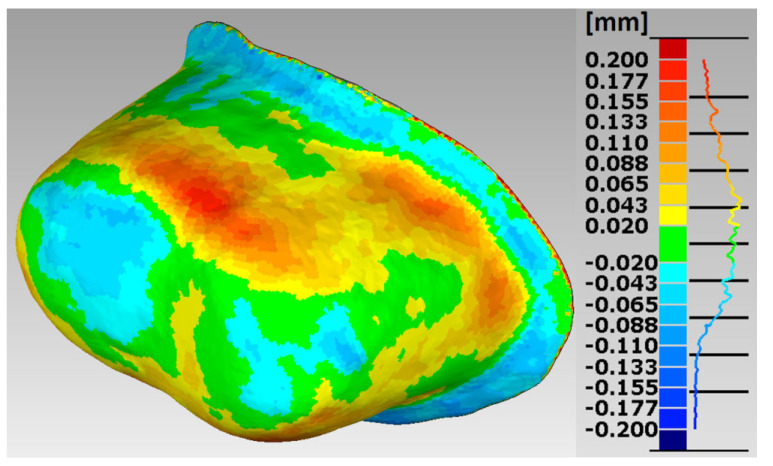
Map of errors in mapping the shape of a crown made of Ti6Al4V alloy in the milling for premolar 15.

**Figure 9 materials-15-05835-f009:**
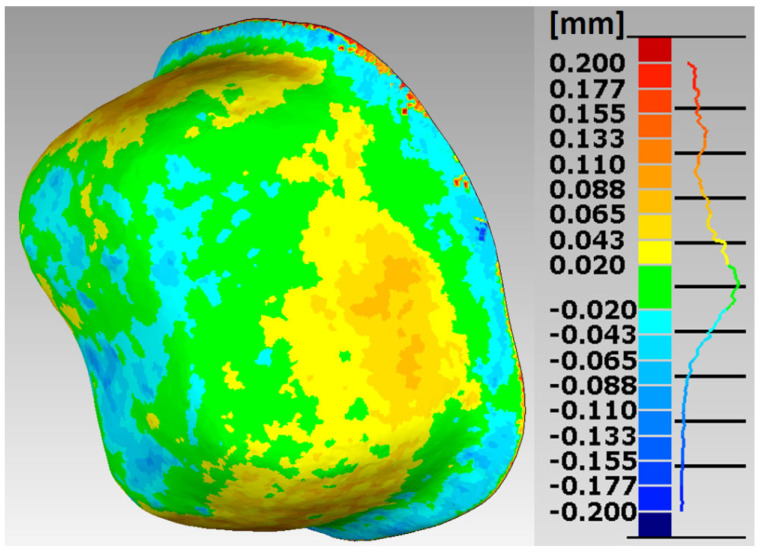
Map of errors in mapping the shape of a crown made of Ti6Al4V alloy in SLM for the molar tooth 36.

**Figure 10 materials-15-05835-f010:**
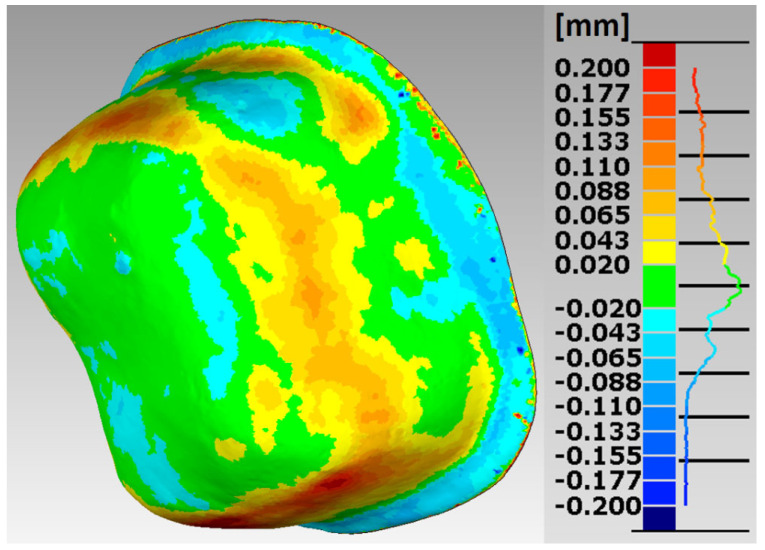
Map of errors in mapping the shape of a crown made of Ti6Al4V alloy in the milling for molar tooth 36.

**Figure 11 materials-15-05835-f011:**
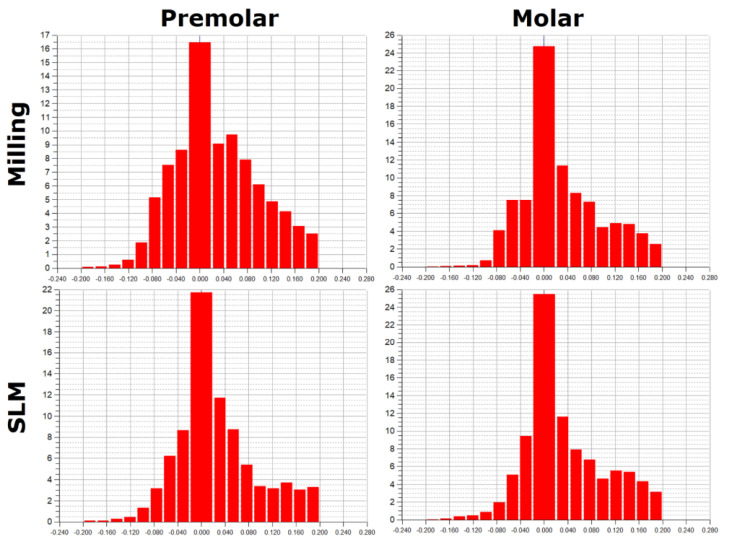
Histograms of the deviation distribution in crowns made of Ti6Al4V alloys for premolars and molars made in SLM and in CAD/CAM milling.

**Figure 12 materials-15-05835-f012:**
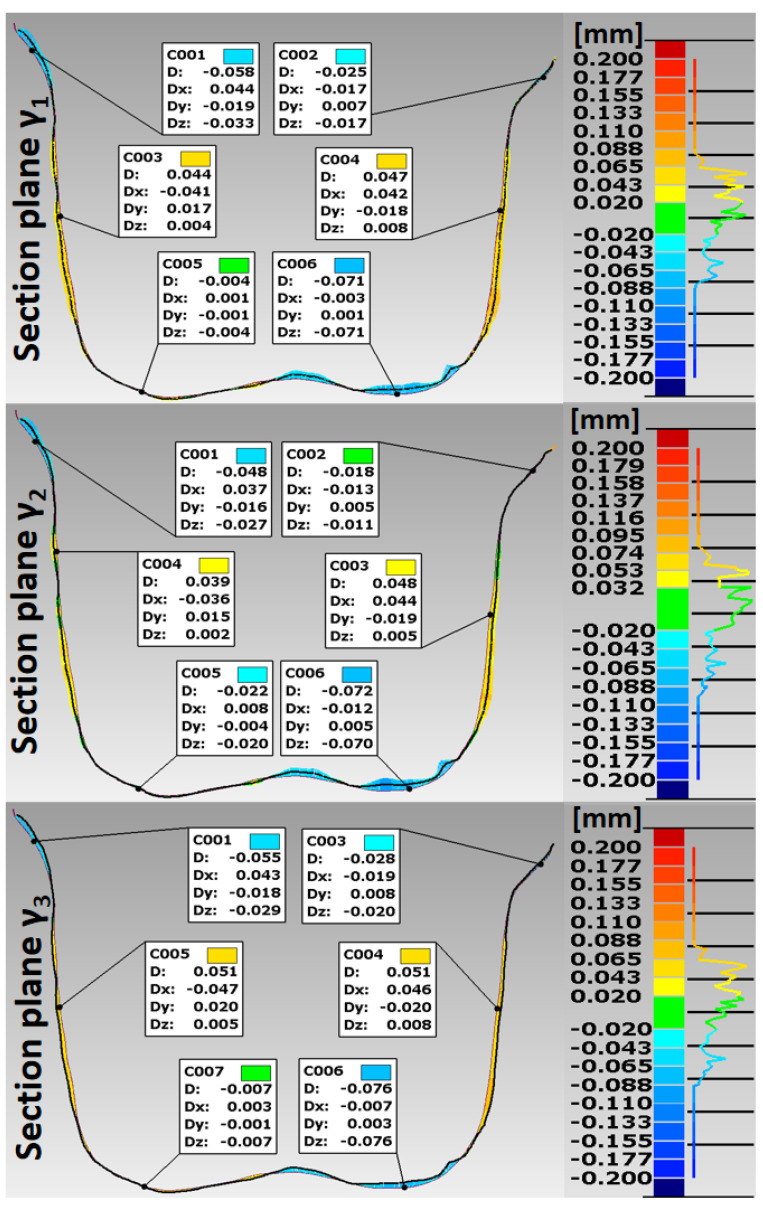
Distribution of deviations in the marginal and internal fit of a crown made of Ti6Al4V alloy in SLM for premolar 15.

**Figure 13 materials-15-05835-f013:**
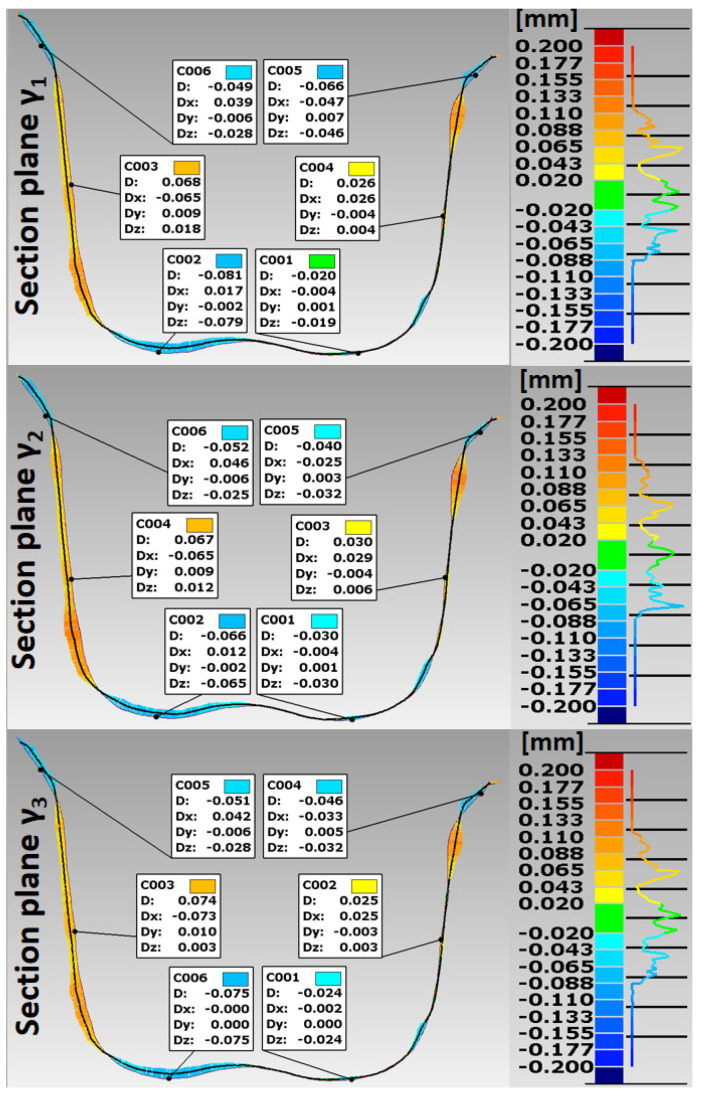
Distribution of deviations in the marginal and internal fit of a crown made of Ti6Al4V alloy in milling for premolar 15.

**Figure 14 materials-15-05835-f014:**
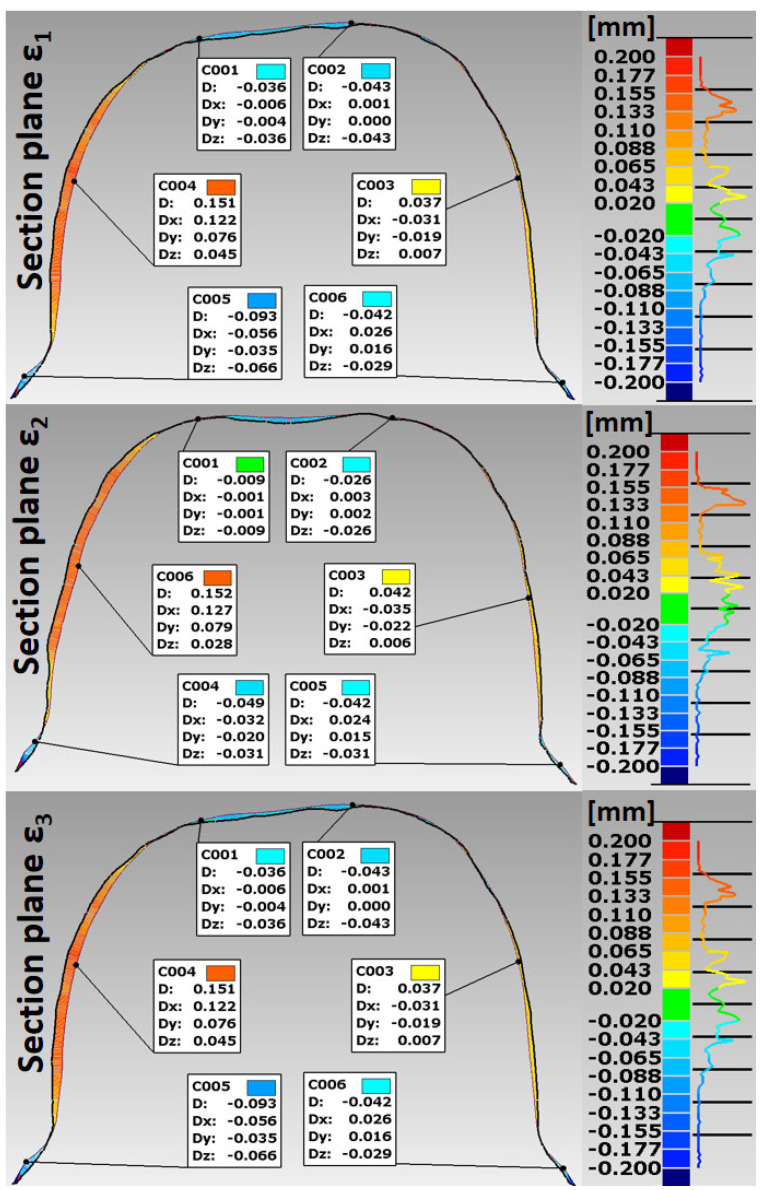
Distribution of deviations in the marginal and internal fit of a crown made of Ti6Al4V alloy in SLM for molar tooth 36.

**Figure 15 materials-15-05835-f015:**
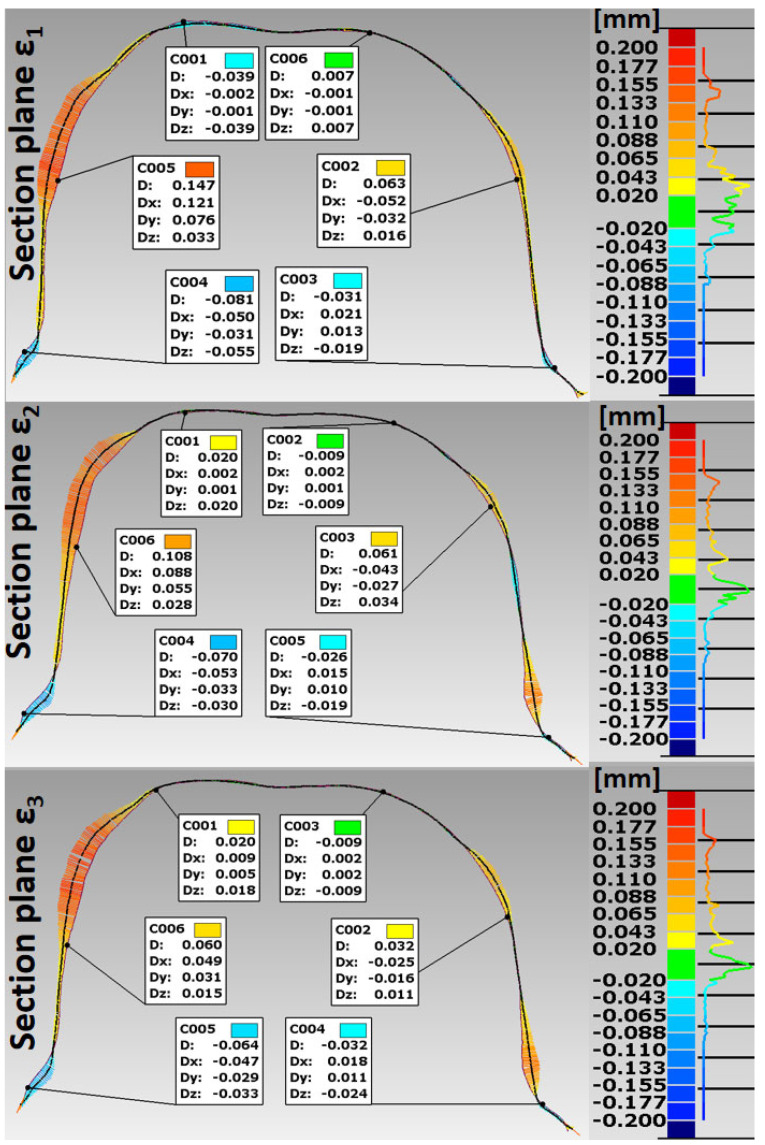
Distribution of deviations in the marginal and internal fit of a crown made of Ti6Al4V alloy in the milling for molar tooth 36.

**Table 1 materials-15-05835-t001:** List of average deviations in the crown fit for premolars and molars in relation to clinically developed abutments.

Technology of Manufacturing Crowns		Premolar 15		Molar 36	
		SLM	Milling	SLM	Milling
**The accuracy deviations of mapping the shape of crowns, mm**	Mean positive deviations	0.107	0.110	0.091	0.096
	Mean negative deviations	−0.039	−0.048	−0.033	−0.038
	Standard deviation	0.068	0.063	0.056	0.061

## Data Availability

The data presented in this study are available on request from the corresponding author. The data are not publicly available due to it is part of a large database and is linked to other clinically proprietary personal data.

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
