# Peer review of "The Impact of Sintering Technology and Milling Technology on Fitting Titanium Crowns to Abutment Teeth—In Vitro Studies"

_materials, 2022, doi:10.3390/ma15175835_

Round 1

Reviewer 1 Report

The aim of the paper is to evaluate the marginal and internal fit of single crown fixed dentures made of Ti6Al4V alloy by SLM and CAD/CAM milling. It is done by new developed method based on digital technologies. Using this method, the accuracy of mapping the shape of crowns in relation to clinical prosthetic abutments can be evaluated and compared.

The manuscript is well structured and the new research method is used which ensures accuracy and reliability of the results. All tables, figures and references are cited in the text. There are no missing references in the list. The abstract is well designed and informative enough. The conclusion corresponds to the obtained results.

There are several remarks:

Abstract:

  1. The SLM abbreviation should be defined.

Introduction:

  1. Page 2, row 54: the sentence „The aim of the study is to compare and evaluate the marginal and internal fit of 54 single-crown fixed dentures made of Ti6Al4V alloy made in two technologies: SLM and 55 CAD/CAM milling.” is gramatically incorrect and not clear enough. It should be edited.
  2. Page 2, rows 71-78: the sentences: „...The crowns ........the working model.” are not clear enogh. The second sentence is too long. Avoid too long sentences. They should be edited.

Discussion:

  1. Page 12, rows 231-233: the sentence “In crowns made of ……. And molar crowns” is gramatically incorrect and not clear enough. It should be edited.
  2. Page 12, rows 233-236: the meaning of the next sentence is opposite to the previous one: „The worse mapping ......... of titanium alloys”. Please check and edit.
  3. Page 12, Fig. 13 is not part of this study. May be it is cited from the reference [17]. If so, there is no need to pu this figure in the manuscript.
  4. Page 13, rows 272-291: The sentences are more suitable for Introduction chapter than for the Discussion.
  5. Page 13, rows 292-301: the new developed method is very well explained in Materials and methods chapter. There is no need to repeat this explanation in Disscussion, moreover at the end the Discussion the special features and advantages of the new method are pointed out one more time.

The English of the whole manuscript should be checked by professional.

Reviewer 2 Report

Dear Authors, this article about the fitting of titanium crowns is really interesting and its clinical impact may be very helpful to clinicians.

Some issues need to be addressed before its publication.

Abstract: please re-write the abstract with headings: introduction, materials and methods, results, conclusion.

Introduction:

Introduction is well written but it is really to short and it is missing some parts that might be helpful to readers. Please add a part about digital impressions.

This reference can help: De Francesco, M.; Stellini, E.; Granata, S.; Mazzoleni, S.; Ludovichetti, F.S.; Monaco, C.; Di Fiore, A. Assessment of Fit on Ten Screw-Retained FrameworksRealized through Digital Full-Arch Implant Impression. Appl. Sci. 202111, 5617. 

Line 55: please write also in words "Ti6Al4V", some readers may not understand

Line 37-40: too many references are being used for this 3 lines sentence. Please write more sentences or erase some references.

Materials and methods:

this part is well written and easily understandable to readers, i would suggest to divide this part into "sub-parts" in order to make it even easier for readers: Example: methods; SEM, Statistical analysis etc.

Results: OK

Discussion: please add a "limitation of the study" section at the end of discussion

Line 237-245: not so easy to understand this methodology, please re-write more clearly

Conslusion: OK

Reviewer 3 Report

Article type:

This article valuated only one material using just a single laboratorial method. This is not enough to be considered a full-length article but instead a communication. Please change it.

Title:

Technology is too  vague. Please specify it.

In addition describe that this is a in-vitro study.

Abstract:

Describe the values of internal fitting per group.

What kind of statistical test was used? Describe it and the p-value.

Introduction:

Insert a reference for the first paragraph.

“Due to the material characteristics of titanium and its alloys (biocompatibility, resistance to corrosion and electrochemical degradation, lightness, low thermal conductivity, low allergic impact, low Young's modulus and high mechanical strength), it is currently the most preferred metal for prosthetic and implant-prosthetic structures” However titanium is not the most suitable material for prosthetic crowns, specially in tooth-supported restorations.

Describe in your introduction when a titanium crown should/can be used, what are the advantages of it instead other restorative materials more esthetical such composites or ceramics.  You need to provide references showing that the clinical condition that you are evaluating is real and the evaluated problem is indeed a real problem.

Insert references with clinical reports/clinical trials that have performed a titanium crown for tooth-supported restoration.

Introduce the difference titanium alloy and why Ti6Al4V should be evaluated.

Methods:

Justify the use of Ti6Al4V,

What kind of abutment material was used? Human tooth, typodont, bovine tooth? How they are standardized? Improve your methods.

What kind of material was used to fix the tooth.

Why have you used spacer varnish? What thickness?

The sample size calculation is missing.

Round 2

Reviewer 3 Report

I am satisfied with the corrections.